# The Role of Corticosteroids in Non-Bacterial and Secondary Encephalitis

**DOI:** 10.3390/life14121699

**Published:** 2024-12-22

**Authors:** Giusy Di Flumeri, Luca Gregorio Giaccari, Maria Caterina Pace, Maria Beatrice Passavanti, Vincenzo Pota, Vincenzo Riccardi, Simona Brunetti, Pasquale Sansone, Francesco Coppolino, Caterina Aurilio

**Affiliations:** 1UOC Emerging Infectious Disease with High Contagiousness, AORN Ospedali dei Colli P.O. C Cotugno, 80131 Naples, Italy; giusydiflumeri@ospedalideicolli.it; 2Department of Woman, Child and General and Specialized Surgery, University of Campania “Luigi Vanvitelli”, 80134 Naples, Italy; lucagregorio.giaccari@gmail.com (L.G.G.); mariacaterina.pace@unicampania.it (M.C.P.); mariabeatrice.passavanti@unicampania.it (M.B.P.); vincenzo.pota@inwind.it (V.P.); vincenzo.riccardi2@studenti.unicampania.it (V.R.); simona.brunetti@studenti.unicampania.it (S.B.); francesco.coppolino1987@gmail.com (F.C.); caterina.aurilio@unicampania.it (C.A.)

**Keywords:** brain inflammation, corticosteroids, dexamethasone, encephalitis, glucocorticoids

## Abstract

Encephalitis affects 1.9 to 14.3 people per 100,000 each year, and the mortality rate varies but can be up to 40%. After the identification of a particular microorganism in a patient with encephalitis, appropriate antimicrobial therapy should be initiated. Corticosteroid therapy represents a therapeutic option in the treatment of primary central nervous system diseases due to its ability to reduce the inflammatory commitment of CNS and consequently reduce mortality rates regardless of the causative agent of injury. Corticosteroid therapy represents a therapeutic option in the treatment of primary central nervous system diseases. Their use is also recommended in meningitis with autoimmune etiology. While corticosteroids have repeatedly been used as adjunctive treatment in encephalitis of viral etiology, the scientific evidence supporting their effectiveness remains scarce. The use of standard doses recommended by the guidelines seems reasonable as an initial setting, especially when a definitive diagnosis of the causal agent is still awaited. The subsequent adjustment should be personalized based on the individual clinical response.

## 1. Introduction

Encephalitis is an inflammation of the brain [1]. There are two main types of encephalitis: primary and secondary. Primary encephalitis occurs when an infectious agent, most commonly a virus, infects the brain. The viruses most involved are herpes simplex virus (HSV) types 1 and 2, varicella zoster virus, enteroviruses, adenovirus, parechovirus, measles virus, and HIV. Secondary encephalitis occurs when an autoimmune response that targets healthy brain cells is triggered by an infection, vaccine, or cancer.

The disease affects 1.9 to 14.3 people per 100,000 each year [2]. Although diagnosis and treatment have improved, the mortality rate varies but can be up to 40% [2]. Of those who survive, those who remain symptomatic have difficulty concentrating, behavioral and speech disturbances, and/or memory loss. In rare cases, patients may remain in a vegetative state [2].

Empirical antimicrobial agents should be initiated on the basis of specific epidemiologic or clinical factors [3]. After the identification of a particular microorganism in a patient with encephalitis, appropriate antimicrobial therapy should be initiated [3].

Researchers have paid attention to the use of corticosteroids as supportive therapy in Central Nervous System (CNS) diseases with primary microbial etiology (bacterial encephalitis and brain abscesses of bacterial, protozoan, and parasitic nature), but not on non-bacterial and secondary encephalitis [4].

Corticosteroid therapy represents a therapeutic option in the treatment of primary central nervous system diseases due to its ability to reduce the inflammatory commitment of CNS and consequently reduce mortality rates regardless of the causative agent of injury [4]. Their use is also recommended in meningitis with autoimmune etiology [4] as first and second-line immunotherapy.

While corticosteroids have repeatedly been used as adjunctive treatment in encephalitis of viral etiology, the scientific evidence supporting their effectiveness remains scarce [4].

Our aim is to review the literature in this regard in order to provide clinicians with further information regarding the advantages and disadvantages of using corticosteroids in non-bacterial and secondary encephalitis.

## 2. Materials and Methods

We conducted a narrative review on the use of corticosteroids in clinical practice as a supportive therapy for non-bacterial and secondary encephalitis.

We identified studies in the main electronic databases. The search strategy was designed for the identification of English language articles published until June 2024 using the following relevant keywords, alone or in combination: (((“Encephalitis, viral”[Mesh]) OR (“Brain Inflammation, viral”[Mesh])) AND ((“Glucocorticoids”[Mesh]) OR (“Corticosteroids”[Mesh]))).

Inclusion criteria encompassed reviews and case–control series; only articles written in English were included. Papers discussing different topics (bacterial encephalitis and autoimmune encephalitis) were excluded. Time or publication status restrictions were not applied. Additional studies were added based on a review of bibliographies of the identified papers.

We focused, in particular, on the advantages and disadvantages that steroid therapy brings to herpetic diseases of the central nervous system in terms of outcome and as an accompanying therapy to structured antibiotic therapy in encephalopathies secondary to bacterial sepsis.

We focused on the class of corticosteroids used, the optimal dosage, and the duration of the therapy itself, and both on the early start of supportive therapy and on a reasoned delay.

## 3. Discussion

### 3.1. Mechanisms of Inflammation of Neuronal Membranes

CNS infections can be divided, for their first engagement site, into primary (acute bacterial meningitis, encephalitis, abscesses, etc.) and secondary ones, usually subsequent to hematogenous spread or originated by contiguity (viral meningitis, abscesses, etc.) [1,2].

There are several types of encephalitis, generally classified into two main categories: infectious encephalitis caused by viruses, bacteria, fungi, or parasites, and autoimmune encephalitis (i.e., non-bacterial NMDAR-associated encephalitis). In addition to these, there are other less common forms, such as post-infectious and limbic encephalitis.

When pathogens pass the blood–brain barrier or replicate consistently in the bloodstream, the human body responds with a systemic inflammatory activation with an increase in serum levels of systemic intravascular cytokines and chemokines inside and outside the barrier [5,6]. This similarly occurs in sepsis, where CNS involvement remains a situation burdened by high mortality despite adequate and timely antibiotic therapy.

Neuroinflammation is defined as an inflammatory process that affects the brain and spinal cord. The course, duration, and intensity of neuroinflammation define the severity of the process and the outcome. The intensity and duration of the process depend on the cascade of inflammatory mediators, such as cytokines, chemokines, and free radicals produced by systemic and endothelial cells (see Figure 1) [7].

The pathophysiological process is sustained by the exposure of the CNS to damage-associated molecular patterns (DAMPs) in response to the direct invasion of the CNS by a pathogen and subsequent immune-mediated encephalitis, resulting in immune system injury to the CNS [8].

The endothelial permeability of the blood–brain barrier is highly susceptible to peripheral pro-inflammatory mediators, altering its permeability [9,10,11]. The resulting neuroinflammatory response causes synaptic impairment and neuronal death [10,11]. Microglia and astrocytes are the major immune cells involved, releasing inflammatory mediators (nitric oxide (NO), prostaglandins, and cytokines), with the subsequent migration of peripheral immune cells into the lesion site. Moreover, pathogens can by themselves cause tissue damage, vasculitis, and tissue ischemia (e.g., *West Nile Virus, Varicella Zoster Virus, Herpes Simplex Virus*) [12,13].

Emerging evidence suggests that a neuroinflammatory state incurred during acute neurological diseases resulting in neuronal damage and dysfunction induces synaptic impairment [14].

Neuroinflammation is only one of the varieties of neurological/neurochemical outcomes resulting from CNS infection, stroke, trauma, or neurodegenerative disorders. They include the breakdown of the blood–brain barrier, glucose deprivation, acidosis, oxidative stress, and excitotoxicity [15].

### 3.2. Neuroinflammation in Non-Bacterial Encephalitis

Central nervous system infections can have three scenarios [16]:Parenchymal infections, which manifest as focal or diffuse encephalitis (primary bacterial and secondary encephalitis, sepsis);Inflammation of the brain secondary to meningeal infections or parameningeal infections (primary bacterial and secondary encephalitis, sepsis);Focal parenchymal infections, surrounded by undamaged brain parenchyma, with a unitary or multifocal character (brain abscess, empyema).

*Herpes simplex virus* (HSV), *herpes zoster virus* (HZV), *cytomegalovirus* (CMV), and *West Nile virus* (WNV) are the most common causes of viral encephalitis. In this kind of infection, the neuronal damage associated with the release of pro-inflammatory cytokines and chemokines is added to the direct viral cytolysis damage [17,18].

Other infections such as Progressive Multifocal Leukoencephalopathy (PML—JC virus related) or Subacute Sclerosing Panencephalitis (measles virus) are also direct manifestations of a neuroinflammatory mechanism, partly indirect, characterized, however, by an insidious and slow progression. They have a long incubation and a slowly progressive course [17,18].

Although severe acute respiratory syndrome coronavirus 2 (*SARS-CoV-2*) predominantly infects the respiratory system, several studies have demonstrated the involvement of the CNS during the disease, of which encephalitis is one of the symptoms [19]. *SARS-CoV-2* infection can cause encephalopathy, probably due to the activated neuroinflammation, but it is not yet clear whether, in part, the neuronal damage can be associated with direct tissue invasion, hypoxia, micro-macro-ischemia, hemorrhages, or other causes [20].

Brain involvement may also be a manifestation of post-infectious mechanisms because infectious pathogens are among the most important environmental triggers in the pathogenesis of autoimmunity [21]. Complex interactions between genetic, immunological, and environmental factors contribute to this loss of tolerance. A variety of pathogens have been implicated in the development of any single autoimmune disorder, suggesting that infections may trigger autoimmunity through mechanisms beyond molecular mimicry. The cumulative exposure to infections during childhood has also been proposed as an important factor in autoimmunity in adulthood [21]. For example, Guillain-Barre syndrome is mainly associated with *Campylobacter jejuni*, *Cytomegalovirus*, *Epstein–Barr virus*, *Influenza*, *HIV*, *Hepatitis A*, *E. Chlamydia pneumonia*, *Mycoplasma pneumonia*, *Hemophilus influenzae*, *Zika virus*, and *SARS-CoV-2* [21]. Group A *Streptococcus* can cause Sydenham’s chorea [21]. Myasthenia gravis could be a consequence of *West Nile Virus*, and neuromyelitis optica spectrum owes its presence mainly to *Varicella Zoster Virus* [21].

Involvement of the CNS is the most severe consequence of some parasitic infections [22]. Of them, protozoal infections include African and American trypanosomiasis, leishmaniasis, malaria, toxoplasmosis, and amoebiasis. Mechanisms underlying invasion of the brain parenchyma by protozoa are not well understood and may depend on the parasite’s nature: a vascular invasion route is most common [22].

Detection of a parasite by the host immune system outside the brain therefore likely triggers a microglial reaction, as has been demonstrated by peripheral injection of the bacterial cell wall component lipopolysaccharide (LPS) [23,24]. LPS, in turn, can induce variable responses of the immune system ranging from mild flu-like symptoms to sepsis [24]. In contrast to bacteria, many parasites have evolved strategies to co-exist with their host. Consequently, many parasitic infections (with notable exceptions according to the virulence factors of determinate strains) are long-lasting diseases that take months or even years to progress [22].

Under such scenarios, peripheral inflammation during the early stages of parasitic infections may be weaker compared with that induced by LPS and would be detected by microglia only in areas to which cytokines diffuse first. Therefore, it is likely that while such parasites are located in the periphery, any microglial reaction would not be widespread but instead be heterogeneous, showing an activated phenotype only in regions to which humoral information is accessible. Upon parasitic invasion of the brain, however, microglial activation would become much more evident [22].

### 3.3. Neuroinflammation in Sepsis and Secondary Encephalitis

Sepsis is a severe clinical syndrome due to the host response to infection, exacerbated by the production of both pro-inflammatory cytokines (TNF-alpha, IL-1-beta>alpha, IL-6, il-8, IL-12, IL-17, COX-2, IRF3) and anti-inflammatory (IL-10) cytokines [25].

Cytokines are a category of relatively small proteins (<40 kDa) produced by leukocytes, macrophages (IL-6, IL-12), dendritic cells (IL-12), and endothelial cells for cell signaling and to promote activation, proliferation, and death of cells. They are divided into interleukins (ILs), chemokines (CKs), interferons, tumor necrosis factors (TNF), and growth factors (GF) [26]. ILs represent the most important family of cytokines released during infection.

The overproduction of cytokines and chemokines leads to an inflammatory storm that increases the severity of the infection; indeed, they represent markers (the chemokine RANTES is inversely associated with APACHE II score) and causative agents of poor outcomes at the same time [27,28,29].

Moreover, cytokine overproduction may be associated with the immune dysregulation observed in sepsis, and an over-activation cascade will rapidly become independent from any invading microorganism stimuli [27,28,29].

The endothelium plays a leading role in inflammatory activation to the extent that pro-inflammatory cytokines promote the expression of adhesion molecules, which allow immune cells to adhere and enter vessel walls and go inside inflamed tissues, inducing ischemia and tissue injury [27,28,29]. NO synthesis at higher levels promotes vasodilation.

Polymorphisms on cytokine genes (i.e., TNF-alpha 1-2 alleles and IL-1Ra 1-2 alleles) have their impact on the course of sepsis, showing an increased severity of the infection in TNF2 carriers and in IL-1Ra2 (associated with higher levels of IL-1 and lower levels of IL-1Ra). G-1082A SNP of the IL-10 has been associated with a more severe form of pneumonitis [30].

### 3.4. Role of Corticosteroids in Viral and Secondary Encephalitis

Corticosteroids are strong anti-inflammatory agents able to reduce the secondary inflammation-mediated damage in encephalitis of various natures when added to a specific antiviral treatment [31]. The beneficial effects of steroid therapy in viral encephalitis are anti-edematous action and possible prevention or therapy of secondary autoimmune phenomena [31].

Steroid use seems to be safe in most cases; no or only minor transient adverse effects are described [31]. There are cases, especially in cases of predisposing factors such as thyroid disorders, in which cerebral venous thrombosis may occur [32]. Corticosteroids have been associated with an increased risk of venous thromboembolism, but it is not clear whether the thrombotic risk is induced by the inflammation of the underlying inflammatory diseases or whether corticosteroids are also prothrombotic [33].

Neuroimaging and lumbar puncture are essential initial diagnostic studies for evaluating patients with viral encephalitis. Lumbar puncture should be performed immediately in patients with a suspected brain infection, and empirical treatment should be started immediately thereafter. Adjunctive corticosteroids are administered to a substantial portion of patients with viral meningitis pending confirmatory testing, such as PCR or culture. Despite the potential concern of increased viral replication with steroids, animal studies have shown no change in herpes simplex virus (HSV) viral load concentrations in cerebrospinal fluid [34]. Steroid therapy can be effective in the acute stage of viral encephalitis.

Three studies report the benefit of steroid therapy in viral encephalitis and have shown, at the same time, few side effects induced by corticosteroids [35,36,37]. No, or only minor, transient adverse effects were reported. The most common diseases reported were HSV1-encephalitis, West Nile, and Measles. Sarkary et al. treated 737 cases of Japanese Encephalitis with a 12 mg daily dose of dexamethasone, comparing the clinical outcome to 462 controls and showing significantly lower mortality in the treatment group and no significant differences in adverse events [35]. Duniewicz et al. treated encephalitis with 5–10 mg/kg/day of hydrocortisone and they found a more rapid normalization of vital parameters, headache, and dizziness, with no differences in adverse events [36]. Kamei et al. point out a better clinical outcome in HSV-1 encephalitis patients treated with dexamethasone or prednisolone at 40–96 mg prednisolone equivalent per day for 2–42 days [37]. Table 1 summarizes the three studies.

This is in contrast to the use of corticosteroids for the treatment of acute bacterial encephalitis. Adjunctive corticosteroids were effective in reducing hearing loss and neurological sequelae in patients with bacterial meningitis caused by all pathogens but did not reduce overall mortality [38,39,40,41,42]. Furthermore, corticosteroids may increase secondary infections or adverse events, such as hyperglycemia, psychosis, or avascular necrosis [42].

### 3.5. The Use of Corticosteroids in the Treatment of Severe Sepsis and Septic Shock

The use of steroid therapy in sepsis still appears controversial today. Already in the 1970s, Schumer demonstrated that low doses of corticosteroids in association with antibiotic therapy during septic shock were associated with a better outcome [43]. In the 1980s Sprung showed a reversal of shock but not an improved overall survival [44].

The purpose of the use of corticosteroids in sepsis is to try to suppress the overproduction of inflammatory mediators, which determines immune decompensation in sepsis up to the patient’s death [45]. For this reason, high doses (30 mg/kg methylprednisolone or equivalent) were used.

In the 1980s, various trials actually demonstrated the lack of impact of the use of corticosteroids on mortality and morbidity in sepsis; therefore, their use in this scenario lost consensus, and then got back on track in the 1990s, with the theory of relative adrenal insufficiency [46,47,48,49,50,51].

The CORTICUS trial showed no benefit on survival [52]. In this large randomized controlled trial, hydrocortisone use did not reduce mortality in the general population of patients with septic shock, although the drug accelerated remission of shock.

Other trials showed some benefit. Annane et al. showed that corticosteroids (low doses of hydrocortisone and fludrocortisone) significantly reduced the risk of death in patients with septic shock and relative adrenal insufficiency without increasing adverse events [51].

All studies consistently showed a decrease in the time of reversal of shock; namely, a decrease in time to the maintenance of adequate blood pressure (defined most commonly as systolic BP > 90 mmHg) without vasopressors.

The use of corticosteroids in sepsis has evolved over the past 40 years. Starting from an initial high dosage and a shorter duration steroid therapy in the 1980s, they have changed over the years, with low doses and longer durations, and are only being used in patients in septic shock who are unresponsive to therapy.

The use of corticosteroids seems to confer a survival benefit only in severely ill septic shock patients—those unresponsive to fluid and vasopressor therapy. The evidence comes from the results of the study by Annane et al. and from the meta-analyses. As reported in the “Surviving Sepsis Campaign Guidelines” updated to 2021, only hydrocortisone (at a dose of 200 mg/die) has been suggested by the SSC for adult septic shock patients not reaching the target MAP despite vasopressor administration [53,54]. In a recent meta-analysis involving over 9000 subjects, Fong et al. showed that a glucocorticoid shortened the time to the resolution of septic shock and the duration of mechanical ventilation while not affecting LOS or mortality [55]. Notably, the combination of a glucocorticoid and fludrocortisone improved short- and longer-term mortality.

More research is required in the field. Unanswered questions remain in the area of less severely ill septic shock patients, timing, tapering, and adverse events. None of the studies had adverse events as a primary endpoint. Furthermore, even amongst the severely ill septic shock patients, for those that the current guidelines recommend treating with corticosteroids, results are inconclusive, and more data are needed.

### 3.6. The Main Corticosteroids Used in CNS Pathologies

Corticosteroids, such as hydrocortisone, dexamethasone, and methylprednisolone, are used for numerous conditions, including neurologic diseases. Corticosteroids have both anti-inflammatory and immunosuppressive properties.

Dexamethasone is a long-acting corticosteroid; its excellent CNS penetration and extensively studied anti-inflammatory properties have been exploited for the treatment of infectious diseases of the brain and spinal cord [56].

In patients with bacterial meningitis, current practice guidelines recommend that dexamethasone (10 mg) should be administered intravenously every 6 h for 4 days [54]. Patients with tuberculous meningitis with a Glasgow Coma Scale (GCS) score < 15 or who have a focal neurological deficit are treated with intravenous dexamethasone for 4 weeks, followed by a taper of oral dexamethasone; patients with a normal mental status and no neurological findings receive intravenous dexamethasone for 2 weeks, followed by the same oral taper as described above [56]. Adjunctive dexamethasone should be used for patients with HSV encephalitis and severe brain edema or vasculitis; however, the use of this agent in these conditions is not supported by systematic evidence [56].

Hydrocortisone may improve the neurological outcomes and mortality rate after traumatic brain injury. It may reduce the rate of hyponatremia and brain swelling. Hydrocortisone could also prevent neuronal apoptosis [57].

Methylprednisolone is used in spinal trauma patients with the aim of mitigating inflammation, lipid peroxidation, and excitotoxicity associated with acute injury [58].

While there is evidence that steroids decrease inflammation, there is still a gap in the literature as to whether they have a defined therapeutic use in the treatment of non-bacterial and secondary encephalitis. Given the significance of this condition, we suggest that corticosteroids should be the subject of dedicated clinical trials to further explore their implications and treatment options.

### 3.7. New Pharmacological Strategies: Non-Steroidal Anti-Inflammatory Drugs (NSAIDs)

NSAIDs have long been used to treat fever and inflammatory diseases. Traditional NSAIDs such as aspirin, ibuprofen, and indomethacin inhibit the activity of both cyclooxygenase (COX) isoforms, COX-1 and COX-2, thereby blocking the production of prostaglandins [59]. The antiviral effects of NSAIDs have been shown to exert an influence on influenza virus, CMV, and VZV. Inhibition of COX activity, scavenging of free radicals, and the down-regulation of transcription can be attributed to the antiviral action of NSAIDs [59].

Advances in the knowledge of the mechanisms involved in COX pathways have led to the consideration of NSAIDs as possible therapeutic options in neuroinflammatory diseases. The expression profile, with COX-1 confined to microglia and COX-2 to neurons, is the background behind the selection of COX-1 inhibitors as effective clinical options against neuroinflammation. Therefore, acetylsalicylate, as an irreversible COX-1 inhibitor, has been found to reduce neuroinflammatory and oxidative injury [60,61].

## 4. Conclusions

The use of steroid therapy in various forms of encephalitis has found clinical benefits in many cases and is associated with a scarcity of adverse events. Its use is expected in association with a causal therapy (antibiotic and/or antiviral) that is rapidly established, adequate in the spectrum of action, respecting the pharmacokinetics and pharmacodynamics of the drug in dosage and duration, and obviously adapting it to the individual clinical response. The use of standard doses recommended by the guidelines seems reasonable as an initial setting, especially when a definitive diagnosis of the causal agent is still awaited. The subsequent adjustment should be personalized based on the individual’s clinical response.

## Figures and Tables

**Figure 1 life-14-01699-f001:**
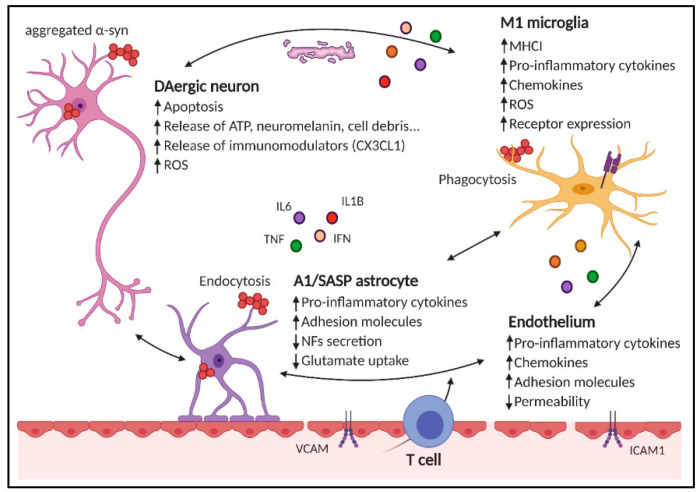
Cascade of inflammatory mediators. Illustration of the inflammatory response produced by oligodendrocytes, astrocytes, and microglia to viral infection. (ATP, adenosine triphosphate; DAergic, dopaminergic; ICAM1, intercellular adhesion molecule 1; MHCI, major histocompatibility complex class I; NF, necrosis factor; ROS, reactive oxygen species; SASP, senescence-associated secretory phenotype; VCAM, vascular cell adhesion molecule).

**Table 1 life-14-01699-t001:** Study summary [35,36,37].

Study	No. of Patients (CS Cases)	CS (Dose)	Viral Agent	Mortality CS vs. non-CS (%)
Sarkary et al. [35]	1282 (737)	Dex (12 mg/day)	JE	42.47 vs. 42.86 (*p* > 0.001)
Duniewicz et al. [36]	138 (109)	Hyd (5–10 mg/kg/day)	Miscellaneous (viral)	?
Kamei et al. [37]	45 (22)	Dex, Pred (40–96 mg prednisolone equivalent per day)	HSV-1	0 vs. 21.7 (*p* < 0.05)

CS, corticosteroids; Dex, dexamethasone; Hyd, Hydrocortisone; HSV-1, human simplex virus 1; JE, Japanese encephalitis; Pred, prednisone; ?, data not available.

## Data Availability

On reasonable request.

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
