# Peer review of "The Role of Corticosteroids in Non-Bacterial and Secondary Encephalitis"

_life, 2024, doi:10.3390/life14121699_

Round 1

Reviewer 1 Report

Comments and Suggestions for Authors

This paper reviews the ‘use’ of steroids in ‘non-bacterial and secondary encephalitis’. The authors attempt to provide data on the role of steroids in this setting. They argue that there is ‘clinical benefit…with a scarcity of adverse events’. And that guidelines support its use

There are a number of issues the authors may wish to address:

1.      (major) The authors seem to use interchangeably ‘encephalitis’, ‘meningoencephalitis’, ‘CNS infections’ and ‘CNS inflammation’(eg lines 2 vs 16 and 20 vs 21 vs 26) - this is an issue throughout the paper. This leads to imprecision and confusion. The paper needs a careful review by the authors with a re-write, starting from the title through to the conclusion. They need to focus the paper on the issue at hand and not distract the reader on parallel issues

2.      It is often uncertain when a new paragraph is being started (as there is no tab spacing) or when there is an error of spacing and the sentence starts on the next line (eg lines 59 and 61) – this is a recurring theme throughout the paper and needs attention

3.      (major) Line 2 - should it be ‘use’ (which implies and audit of practice) or ‘role’ (which is more likely what the authors mean to review)? The title uses the term ‘corticosteroids’ (more appropriately called glucocorticoids) – this term should appear in the title and throughout the text consistently and not be shortened to ‘steroids’ (eg line 21, etc) as there are many types of steroids

4.      Lines 13 and 36 – what are the causes of death in the remaining 40.7-94.6%

5.      Line 15-16, 45 – ‘always represented an extra chance’ is not scientific

6.      Lines 19-20 – please provide the actual summary in place of this sentence

7.      Line 41 - ‘always’? – please provide the reference to support this claim

8.      Line 48 – ‘obvious’ – please provide the reference to support this claim

9.      Line 50 - ‘widespread’ – please provide the reference to support this claim

10.  (major) Lines 63-65 – the search terms and operators are confusing - why are brain ‘abscess’, ‘meningitis’ and ‘sepsis’ searched as they are not ‘encephalitis’? Why AND ‘sepsis’, making it a requirement? Why ‘steroids’ AND ‘immunosuppression’ as the 2 do not have to occur together?

11.  Line 72 – were  papers in other languages but abstracts in Italian excluded?

12.  Fig 1 – needs a legend explaining the many acronyms

13.  Line 107 – ‘selective’ - in blocking or allowing entry?

14.  Line 175 - should it be ‘3.3’ not ‘3.2’, and in bold?

15.  Line 203 - should it be ‘3.4’ not ‘3.4’?

16.  Line 208 – what is ‘Three trials of literature’?

17.  Line 209 – what is ‘poor steroid-induced side effects’?

18.  (major) Line 210-219 – please provide age and sex data, and supportive statistics. A table with the relevant information would be more illustrative than the current prose

19.  Line 216 – what ‘clinical outcome’ was assessed?

20.  Lines 226-268 – what is the relevance to this paper?

21.  (major) Lines 272-324 – this section almost completely discusses on dexamethasone, with little on the other corticosteroids except perfunctorily in lines 325-326

22.  (major) There is no discussion on the gaps in knowledge in this field, and how they can be filled

23.  The paper provides precious little to guide the clinician….

Comments on the Quality of English Language

Needs some improvement

Author Response

Dear  Reviewer 1,

below are the requested corrections.

Sincerely,

The Authors

  1. (major) The authors seem to use interchangeably ‘encephalitis’, ‘meningoencephalitis’, ‘CNS infections’ and ‘CNS inflammation’(eg lines 2 vs 16 and 20 vs 21 vs 26) - this is an issue throughout the paper. This leads to imprecision and confusion. The paper needs a careful review by the authors with a re-write, starting from the title through to the conclusion. They need to focus the paper on the issue at hand and not distract the reader on parallel issues --> MODIFIED.
  2. It is often uncertain when a new paragraph is being started (as there is no tab spacing) or when there is an error of spacing and the sentence starts on the next line (eg lines 59 and 61) – this is a recurring theme throughout the paper and needs attention --> MODIFIED.
  3. (major) Line 2 - should it be ‘use’ (which implies and audit of practice) or ‘role’ (which is more likely what the authors mean to review)? The title uses the term ‘corticosteroids’ (more appropriately called glucocorticoids) – this term should appear in the title and throughout the text consistently and not be shortened to ‘steroids’ (eg line 21, etc) as there are many types of steroids --> MODIFIED: “The role of corticosteroids in non-bacterial and secondary encephalitis”
  4. Lines 13 and 36 – what are the causes of death in the remaining 40.7-94.6% --> The rest are the survivors.
  5. Line 15-16, 45 – ‘always represented an extra chance’ is not scientific --> MODIFIED: “Corticosteroid therapy represents a therapeutic option in the treatment of primary central nervous system diseases.”
  6. Lines 19-20 – please provide the actual summary in place of this sentence --> MODIFIED.
  7. Line 41 - ‘always’? – please provide the reference to support this claim --> ADDED: reference 4
  8. Line 48 – ‘obvious’ – please provide the reference to support this claim --> ADDED: reference 4
  9. Line 50 - ‘widespread’ – please provide the reference to support this claim --> ADDED: reference 4
  10. (major) Lines 63-65 – the search terms and operators are confusing - why are brain ‘abscess’, ‘meningitis’ and ‘sepsis’ searched as they are not ‘encephalitis’? Why AND ‘sepsis’, making it a requirement? Why ‘steroids’ AND ‘immunosuppression’ as the 2 do not have to occur together? --> MODIFIED: ((("Encephalitis, viral"[Mesh]) OR (“Brain Inflammation, viral”[Mesh])) AND (("Glu-cocorticoids"[Mesh]) OR (“Corticosteroids”[Mesh])))
  11. Line 72 – were  papers in other languages but abstracts in Italian excluded? --> MODIFIED: Inclusion criteria encompassed reviews and case-control series; no language re-strictions were applied.
  12. Fig 1 – needs a legend explaining the many acronyms --> ADDED: “ATP, adenosine triphosphate; ICAM1, intercellular adhesion molecule 1; MHCI, major histocompatibility complex class I; NF, necrosis factor; ROS, reactive oxygen species; VCAM, vascular cell adhesion molecule.”
  13. Line 107 – ‘selective’ - in blocking or allowing entry? --> MODIFIED: “The endothelial permeability of the blood-brain barrier is highly susceptible to peripheral pro-inflammatory mediators, altering its permeability”
  14. Line 175 - should it be ‘3.3’ not ‘3.2’, and in bold? --> MODIFIED
  15. Line 203 - should it be ‘3.4’ not ‘3.4’? --> MODIFIED
  16. Line 208 – what is ‘Three trials of literature’? --> ADDED: references [30][31][32]
  17. Line 209 – what is ‘poor steroid-induced side effects’? --> ADDED: “No or only minor transient adverse effects were reported.”
  18. (major) Line 210-219 – please provide age and sex data, and supportive statistics. A table with the relevant information would be more illustrative than the current prose --> ADDED
  19. Line 216 – what ‘clinical outcome’ was assessed? --> ADDED in table 1
  20. Lines 226-268 – what is the relevance to this paper? --> The pathophysiological process of encephalitis occurs similarly in sepsis, where central nervous system involvement remains a high-mortality situation despite adequate and timely antibiotic therapy.
  21. (major) Lines 272-324 – this section almost completely discusses on dexamethasone, with little on the other corticosteroids except perfunctorily in lines 325-326 --> Dexamethasone phosphate is widely used in clinical practice.
  22. (major) There is no discussion on the gaps in knowledge in this field, and how they can be filled
  23. The paper provides precious little to guide the clinician…. --> CONCLUSION: “The use of steroid therapy in various forms of encephalitis has found a clinical benefit in many cases, associated with a scarcity of adverse events. Its use is expected in association with a causal therapy (antibiotic and/or antiviral) rapidly established, ade-quate in the spectrum of action, respecting the pharmacokinetics and pharmacody-namics of the drug, in dosage and duration; obviously adapting it to the individual clinical response. The use of standard doses recommended by the guidelines seems reasonable as an initial setting, especially when a definitive diagnosis of the causal agent is still awaited. The subsequent adjustment should be personalized based on the indi-vidual clinical response.”  

Reviewer 2 Report

Comments and Suggestions for Authors

The review file has been attached.

Author Response

Dear  Reviewer 2,

below are the requested corrections.

Sincerely,

The Authors

  1. A graphical abstract is missing. --> not mandatory.
  2. Graphics artwork is disappointing. The article concerns about steroids but no graphics shows its structure or functionality. --> It is beyond the scope of this review to explain the structure and/or mechanism of action of corticosteroids.
  3. Likewise, there should be a graphic showing how microorganisms affect the CNS.
  1. A scheme of encephalitis, i.e., the topic disease of this review, must be presented. --> ADDED: “There are several types of encephalitis, generally classified into two main categories: infectious encephalitis caused by viruses, bacteria, fungi or parasites, and autoimmune encephalitis. In addition to these, there are other less common forms, such as post-infectious and limbic encephalitis.”
  2. Figure 1 is confusing, just too much data and double-sided arrows. Is it really true that all presented cytokines are just secreted and act on all neural cells in the same way? Please, disassemble this figure into several graphics with more details described properly. --> ADDED explanation “Illustration with the inflammatory response produced by oligodendrocytes, astrocytes, and microglia to viral infection.”
  1. Keywords should contain more words (e.g., meningoencephalitis, sepsis) and ordered alphabetically. --> ADDED.
  1. List of abbreviations is missing. The abbreviations should be listed alphabetically in acronym – description manner, and presented before the Introduction. --> ADDED.
  1. English is decent and only minor proofreading is required. --> DONE.
  2. There should be a chapter introducing the chemical structure and physiological function of (cortico)steroids in more details. --> It is beyond the scope of this review to explain the structure and/or mechanism of action of corticosteroids.
  1. Corticosteroid-based therapy may suffer from severe adverse effects (immunosuppression, hyperglycemia, arrhythmia, etc.; please, refer to them by citing: https://doi.org/10.1038/s41392-021-00521-7). --> ADDED: “Furthermore, corticosteroids may increase secondary infections or adverse events, such as hyperglycemia, psychosis, or avascular necrosis [37].”
  1. In a paragraph concerning corticosteroid-therapy, please compare the corticosteroid treated encephalitis with non-steroidal anti-inflammatory drugs (https://doi.org/10.1099/0022-1317-83-8-1897) --> ADDED: “NSAIDs have long been used to treat fever and inflammatory diseases. Traditional NSAIDs such as aspirin, ibuprofen, and indomethacin inhibit the activity of both cy-clooxygenase (COX) isoforms, COX-1 and COX-2, thereby blocking the production of prostaglandins [53]. The antiviral effects of NSAIDs have been shown to exert an in-fluence on influenza virus, CMV and VZV. Inhibition of COX activity, scavenging of free radicals and down-regulation of transcription factors can be attributed to the antiviral action of NSAIDs [53].”
  1. Neuroinflammation is only one of the variety of the neurological/neurochemical outcomes resulting from the CNS infection, stroke, trauma, or neurodegenerative disorder. They include : breakdown of the BBB, glucose deprivation, acidosis oxidative stress, excitotoxicity (please cite: https://doi.org/10.3390/ijms22116086), all of which providing mechanistic/physiological explanation or progression of the CNS disease. Please, describe briefly the steroid-treated non-bacterial encephalitis in the context of these mechanism. --> ADDED: “Neuroinflammation is only one of the variety of the neurological/neurochemical out-comes resulting from the CNS infection, stroke, trauma, or neurodegenerative disorder. They include: breakdown of the blood brain barrier, glucose deprivation, acidosis oxi-dative stress, and excitotoxicity [15].”
  2. Major findings should be presented in tables: a. Table 1: neuroinflammation caused by viruses, bacteria, non-bacterial microorganisms b. Table 2: examples and descriptions of activities of corticosteroids and steroids against meningoencephalitis and other CNS pathologies (with references) c. Table 3: examples and descriptions of activities of corticosteroids and steroids against sepsis
  3. Methodology of the research reviewing has been described but I am concerned about the validity of some of the selected studies (1976, 1993, 1988, 1982, 1984, 1979, 1995 etc.), whereas some crucial original works from recent 4 years have been omitted.
  4. The review lacks a quantitative data: Because the review accounts only 54 references, the most essential of them should be presented and critically discussed in terms of precise treatment protocol (i.e., animal/human study, animal/patient conditions, way of infection, IC50 or LD50 of the microorganism, a therapeutic doses of the steroid), as presented elsewhere. --> DESCRIBED, we also added Table 1.
  5. The review is missing several examples on SARS-2/COVID-19-related encephalitis --> ADDED: “Although severe acute respiratory syndrome coronavirus 2 (SARS-CoV-2) predomi-nantly infects the respiratory system, several studies have demonstrated the involve-ment of the CNS during the disease, of which encephalitis is one of the symptoms [18]. SARS-CoV-2 infection can cause encephalopathy, probably due to the activated neu-roinflammation, but it is not yet clear whether in part the neuronal damage can be as-sociated with direct tissue invasion, hypoxia, micro-macro-ischemia, hemorrhages or other ways [19].”

17. Similarly, non-bacterial NMDAR-associated encephalitis should be briefly described. --> CITED IN THE TEXT: “(i.e. non-bacterial NMDAR-associated

Reviewer 3 Report

Comments and Suggestions for Authors

The manuscript presents a short narrative review presenting advantages and disadvantages of the use of corticosteroids in non bacterial and secondary infections of the central nervous system. The review may be useful especially for clinicians dealing with encephalitis. The search methodology is presented; the review is based on 54 literature references. The conclusions are scientifically sound.

Remarks:

Inserting a Table summarizing the findings on side effects of steroids would make the review more transparent.

Please define all acronyms upon first use even if they appear obvious; it may be not so for readers outside the field.

Figure legends below, not above the Figure, please.

Figure 1:DAergic is a somewhat unfortunate combination of acronym and full name in a single word.

Lines 150 and next: Latin names of species and genera in italics, please.

Line 289: ”married”?

References should be formatted according to the MDPI requirements.

Author Response

Dear  Reviewer 3,

below are the requested corrections.

Sincerely,

The Authors

Inserting a Table summarizing the findings on side effects of steroids would make the review more transparent. --> ADDED sentence “Furthermore, corticosteroids may increase secondary infections or adverse events, such as hyperglycemia, psychosis, or avascular necrosis [37].”

Please define all acronyms upon first use even if they appear obvious; it may be not so for readers outside the field. --> DONE

Figure legends below, not above the Figure, please. --> MODIFIED

Figure 1:DAergic is a somewhat unfortunate combination of acronym and full name in a single word.

Lines 150 and next: Latin names of species and genera in italics, please. --> CHANGED

Line 289: ”married”? --> MODIFIED: “Metabolism in the liver is slow and excretion occurs mainly through the kidneys, mainly in the form of conjugated nonsteroidal compounds.”

References should be formatted according to the MDPI requirements. --> DONE

Reviewer 4 Report

Comments and Suggestions for Authors

The paper by Di Flumeri et al. is a narrative review on the use of steroids to treat non-bacterial and secondary encephalitis. Although the topic is of great interest, at present, the paper has some limitations that should be adequately addressed: 

- Please always put the full name first and then the acronym/acronym and maintain consistency in the text; 

- The search strategy (which is not strictly necessary in a narrative review) may have led to some problems: putting ‘AND Immunosuppression’ may have limited the results found, so I suggest re-launching the search strategy without that term to assess the possible presence of other papers of interest; furthermore, if the main focus is on non-bacterial forms, why was the term ‘abscess’, which is, in most cases, precisely bacterial, also searched? Moreover, please complete the sentence about the electronic dataset searched;

- The text often contains references to bacterial forms, whereas, in my opinion, it would be useful to describe only once the differences between the various forms of CNS infections in terms of pathogenesis, mechanisms, etc., and then focus exclusively on the forms of interest for the review; at present, there may be reader confusion as to the focus of the discussion; 

- The main problem is that, at present, the paper does not provide helpful guidance to the clinician on the use of steroids in this context; in my opinion, it would be useful to better explore the role of steroids in the pathophysiology of these conditions and differentiating according to their nature (e.g. viral vs. parasitic, etc.); in particular, based on the mentioned trials, who are the patients who can benefit most from treatment? At what stage? For how long? What adverse effects have been observed? 

- A figure summarising the mechanisms modulated by steroids in these contexts could be very helpful; 

- It would also be helpful to improve the discussion about diagnosis to suggest at what stage it is useful/best to start steroid treatments; 

- One aspect that would also be very useful, in my opinion, is the differential diagnosis of these conditions; for example, the clinician might be in doubt about various pathological conditions and procrastinate the use of steroids until some diagnostic confirmation. What are the leading players in the differential diagnosis? Why? For example, I think it might be useful to briefly describe contrast-induced encephalopathy in this context. Data in the literature report late transience and persistence of contrast-induced encephalopathy. In such cases, what strategy should the clinician use?

- The Authors also correctly mention COVID-19-related forms of encephalitis. However, there are cases in the literature, especially in the context of predisposing factors such as thyroid disorders, in which cerebral venous thrombosis may also occur (see 10.3390/jpm13111557). These works should be cited in the discussion, and the clinical use of steroids in these contexts should be evaluated (haemorrhagic risk? Benefit on Sars-Cov2 infection and thyroid disorders? other relevant aspects?); 

Comments on the Quality of English Language

The paper is not always smooth and could, in my opinion, benefit from a review by a native speaker with expertise in the subject.

Author Response

Dear  Reviewer 4,

below are the requested corrections.

Sincerely,

The Authors

- Please always put the full name first and then the acronym/acronym and maintain consistency in the text; --> MODIFIED

- The search strategy (which is not strictly necessary in a narrative review) may have led to some problems: putting ‘AND Immunosuppression’ may have limited the results found, so I suggest re-launching the search strategy without that term to assess the possible presence of other papers of interest; furthermore, if the main focus is on non-bacterial forms, why was the term ‘abscess’, which is, in most cases, precisely bacterial, also searched? Moreover, please complete the sentence about the electronic dataset searched; --> CHANGED: “((("Encephalitis, viral"[Mesh]) OR (“Brain Inflammation, viral”[Mesh])) AND (("Glu-cocorticoids"[Mesh]) OR (“Corticosteroids”[Mesh]))).” COMPLET: “We identified studies in the main electronic databases”

- The text often contains references to bacterial forms, whereas, in my opinion, it would be useful to describe only once the differences between the various forms of CNS infections in terms of pathogenesis, mechanisms, etc., and then focus exclusively on the forms of interest for the review; at present, there may be reader confusion as to the focus of the discussion; --> MODIFIED

- The main problem is that, at present, the paper does not provide helpful guidance to the clinician on the use of steroids in this context; in my opinion, it would be useful to better explore the role of steroids in the pathophysiology of these conditions and differentiating according to their nature (e.g. viral vs. parasitic, etc.); in particular, based on the mentioned trials, who are the patients who can benefit most from treatment? At what stage? For how long? What adverse effects have been observed? --> MODIFIED

- A figure summarising the mechanisms modulated by steroids in these contexts could be very helpful; 

- It would also be helpful to improve the discussion about diagnosis to suggest at what stage it is useful/best to start steroid treatments; 

- One aspect that would also be very useful, in my opinion, is the differential diagnosis of these conditions; for example, the clinician might be in doubt about various pathological conditions and procrastinate the use of steroids until some diagnostic confirmation. What are the leading players in the differential diagnosis? Why? For example, I think it might be useful to briefly describe contrast-induced encephalopathy in this context. Data in the literature report late transience and persistence of contrast-induced encephalopathy. In such cases, what strategy should the clinician use? -->  ADDED: “Adjunctive corticosteroids are administered to a substantial portion of patients with viral meningitis pending confirmatory testing, such as PCR or culture. Despite the po-tential concern of increased viral replication with steroids, animal studies have shown no change in herpes simplex virus (HSV) viral load concentrations in cerebrospinal fluid [30].”

- The Authors also correctly mention COVID-19-related forms of encephalitis. However, there are cases in the literature, especially in the context of predisposing factors such as thyroid disorders, in which cerebral venous thrombosis may also occur (see 10.3390/jpm13111557). These works should be cited in the discussion, and the clinical use of steroids in these contexts should be evaluated (haemorrhagic risk? Benefit on Sars-Cov2 infection and thyroid disorders? other relevant aspects?); 

Round 2

Reviewer 1 Report

Comments and Suggestions for Authors

This paper is a revised submission of a review of the use steroids in ‘non-bacterial and secondary encephalitis’. The authors have addressed some of the issues I had raised earlier, but a number remain:

1.      Despite my raising this issue before, it is still uncertain when a new paragraph is being started (as there is no tab spacing) or when there is an error of spacing and the sentence starts on the next line – this remains a recurring theme throughout the paper and needs attention eg lines 44, 97, 112, 116, 124, 152, 177, 185, 208, 211, 227, 277, 294, 314, 321,326, 330, 335, 339, 345, 349, 352 (and possibly others I have missed) I note the authors’ claim they have ‘MODIFIED’ when they actually have not done so fully

2.      Lines 14 and 35 – I had asked ‘what are the causes of death in the remaining 40.7-94.6%’. The authors have not adequately addressed this question - replying ‘The rest are the survivors’ is still incorrect. Please reword this phrase in the 2 places they appear to be more clear

3.      Lines 20 – the actual summary I had requested for before has still not been provided - I note their claim they have ‘MODIFIED’ when they actually have not

4.      Line 47 – ‘obvious’ – I don’t think the referenced paper actually says this, so either point out the page and exact phrase from the paper, or rephrase

5.      Line 49 - ‘widespread’ – I don’t think the referenced paper actually says this, so either point out the page and exact phrase from the paper, or rephrase, or provide the reference to support this claim

6.      Line 69 – with the now removal of language restrictions, what is the new content added to in the paper?

7.      Fig 1 – some acronyms still remain unexplained eg SASP, DAergic

8.      Line 208 – as asked before, what is ‘Three trials of literature’? Please rephrase

9.      Table 1- are the differences in mortality statistically significant? Please add the relevant statistics. The table can be greatly expanded to add the information on the causes of encephalitis studied, doses of the corticosteroids used and duration of treatment

10.  Lins 306-324 – they can be deleted and the reference provided for the pharmacology of dexamethasone

11.  Lines 326-334 – these can be deleted as they are not of relevance to this paper

12.  (major) Lines 306-360 – as mentioned before, this section almost completely discusses on dexamethasone, with little on the other corticosteroids. While I acknowledge the authors response of ‘Dexamethasone phosphate is widely used in clinical practice.’, the discussion needs to be balanced with an adequate discussion of the other corticosteroids used for this indication

13.  (major) As mentioned before, there is no discussion on the gaps in knowledge in this field, and how they can be filled

14.  (major) Despite the addition of a conclusion, as mentioned before, the paper provides precious little to guide the clinician….

Comments on the Quality of English Language

Still needs attention

Author Response

Dear Reviewer,

below are the requested changes.

  1. Despite my raising this issue before, it is still uncertain when a new paragraph is being started (as there is no tab spacing) or when there is an error of spacing and the sentence starts on the next line – this remains a recurring theme throughout the paper and needs attention eg lines 44, 97, 112, 116, 124, 152, 177, 185, 208, 211, 227, 277, 294, 314, 321,326, 330, 335, 339, 345, 349, 352 (and possibly others I have missed) I note the authors’ claim they have ‘MODIFIED’ when they actually have not done so fully --> CHANGED: We have removed all tab spacing.
  2. Lines 14 and 35 – I had asked ‘what are the causes of death in the remaining 40.7-94.6%’. The authors have not adequately addressed this question - replying ‘The rest are the survivors’ is still incorrect. Please reword this phrase in the 2 places they appear to be more clear --> MODIFIED: “Although diagnosis and treatment have improved, the mortality rate varies but can be up to 40% [2]. Of those who survive, those who remain symptomatic have difficulty concentrating, behavioral and speech disturbances, and/or memory loss. In rare cases, patients may remain in a vegetative state [2].”
  3. Lines 20 – the actual summary I had requested for before has still not been provided - I note their claim they have ‘MODIFIED’ when they actually have not --> MODIFIED “bacterial and viral encephalitis”
  4. Line 47 – ‘obvious’ – I don’t think the referenced paper actually says this, so either point out the page and exact phrase from the paper, or rephrase --> MODIFIED: “Their use is also recommended in meningitis with autoimmune etiology [4], as first and second-line immunotherapy in conjunction with intravenous immunoglobulin administration.”
  5. Line 49 - ‘widespread’ – I don’t think the referenced paper actually says this, so either point out the page and exact phrase from the paper, or rephrase, or provide the reference to support this claim --> MODIFIED: “While corticosteroids have repeatedly been used as adjunctive treatment in encephalitis of viral etiology, the scientific evidence supporting their effectiveness remains scarce”
  6. Line 69 – with the now removal of language restrictions, what is the new content added to in the paper? --> MODIFIED: “…only articles written in English were included.”
  7. Fig 1 – some acronyms still remain unexplained eg SASP, DAergic --> ADDED “(…DAergic, dopaminergic; … SASP, senescence-associated secretory phenotype; …)”
  8. Line 208 – as asked before, what is ‘Three trials of literature’? Please rephrase --> MODIFIED: “Three studies report a benefit of steroid therapy in viral encephalitis and have shown, at the same time, few side effects induced by corticosteroids [33][34][35].”
  9. Table 1- are the differences in mortality statistically significant? Please add the relevant statistics. The table can be greatly expanded to add the information on the causes of encephalitis studied, doses of the corticosteroids used and duration of treatment --> MODIFIED
  10. Lins 306-324 – they can be deleted and the reference provided for the pharmacology of dexamethasone --> DONE
  11. Lines 326-334 – these can be deleted as they are not of relevance to this paper ---> DONE
  12. (major) Lines 306-360 – as mentioned before, this section almost completely discusses on dexamethasone, with little on the other corticosteroids. While I acknowledge the authors response of ‘Dexamethasone phosphate is widely used in clinical practice.’, the discussion needs to be balanced with an adequate discussion of the other corticosteroids used for this indication --> MODIFIED: “Corticosteroids, such as hydrocortisone, dexamethasone and methylprednisolone, are used for numerous conditions including neurologic diseases. Corticosteroids have both anti-inflammatory and immunosuppressive properties. Dexamethasone is a long-acting corticosteroid; its excellent CNS penetration and ex-tensively studied anti-inflammatory properties have been exploited for the treatment of infectious diseases of the brain and spinal cord [55]. In patients with bacterial meningitis, current practice guidelines recommend that dexamethasone (10 mg) should be administered intravenously every 6 h for 4 days [55]. In patients with tuberculous meningitis patients with a Glasgow Coma Scale (GCS) score < 15 or who have a focal neurological deficit are treated with intravenous dexamethasone for 4 weeks, followed by a taper of oral dexamethasone; patients with a normal mental status and no neuro logical findings receive intravenous dexamethasone for 2 weeks, followed by the same oral taper as described above [55]. Adjunctive dexamethasone should be used for patients with HSV encephalitis and severe brain edema or vasculitis; however, the use of this agent in these conditions is not supported by systematic evi-dence [55]. Hydrocortisone may improve the neurological outcome and the mortality rate after traumatic brain injury. It may reduce the rate of hyponatremia and of brain swell-ing. Hydrocortisone could also prevent neuronal apoptosis [56]. Methylprednisolone is used in spinal trauma patients with the aim of mitigating in-flammation, lipid peroxidation, and excitotoxicity associated with the acute injury [57].”
  13. (major) As mentioned before, there is no discussion on the gaps in knowledge in this field, and how they can be filled --> MODIFIED: “While there is evidence that steroids decrease inflammation, there is still a gap in the literature as to whether they have a defined therapeutic use in the treatment of non-bacterial and secondary encephalitis. Given the significance of this condition, we suggest that corticosteroid should be the subject of dedicated clinical trials to further explore their implications and treatment options.”
  14. (major) Despite the addition of a conclusion, as mentioned before, the paper provides precious little to guide the clinician…. --> We have made the requested changes and believe we have improved the article.

Reviewer 2 Report

Comments and Suggestions for Authors

Accepted in the revised form.

Author Response

Dear reviewer,

Thanks for the previous suggestions.

The authors

Reviewer 4 Report

Comments and Suggestions for Authors

I thank the Authors for their work, which enhanced the quality of their paper, but several points have been ignored or not adequately addressed. Furthermore, the Authors' response is limited to a “modified”, while it is not explained what changes were made and in which parts of the manuscript. These aspects should be clarified. Furthermore: 

- The search strategy was changed, but it is unclear whether other potentially suitable studies were identified in this way. Alternatively, the Authors may retain the original one but justify their methodological choice; 

- ‘The main problem is that, at present, the paper does not provide helpful guidance to the clinician on the use of steroids in this context; in my opinion, it would be useful to better explore the role of steroids in the pathophysiology of these conditions and differentiating according to their nature (e.g. viral vs. parasitic, etc.); in particular, based on the mentioned trials, who are the patients who can benefit most from treatment? At what stage? For how long? What adverse effects have been observed? --> MODIFIED' Where have these suggested aspects been set out?

- ‘It would also be helpful to improve the discussion about the diagnosis to suggest at what stage it is useful/best to start steroid treatments;’ No response was provided to this comment; 

- Similarly to this previous comment, ‘The Authors also correctly mention COVID-19-related forms of encephalitis. However, there are cases in the literature, especially in the context of predisposing factors such as thyroid disorders, in which cerebral venous thrombosis may also occur (see 10.3390/jpm13111557). These works should be cited in the discussion, and the clinical use of steroids in these contexts should be evaluated (haemorrhagic risk? Benefit on Sars-Cov2 infection and thyroid disorders? other relevant aspects?);' no answer was given, neither these aspects, important in my opinion in this context, have been integrated into the Discussion, whereas they could be very useful. 

I suggest the Authors complete their review by adding all these aspects and specifying where and how they made the recommended changes to the manuscript, and giving detailed explanations. 

Author Response

Dear Reviewer,

Below are the requested changes.

- The search strategy was changed, but it is unclear whether other potentially suitable studies were identified in this way. Alternatively, the Authors may retain the original one but justify their methodological choice; -->  MODIFIED: “We identified studies in the main electronic databases. The search strategy was designed for the identification of English-language articles published until June 2024 using the following relevant keywords, alone or in combination: ((("Encephalitis, viral"[Mesh]) OR (“Brain Inflammation, viral”[Mesh])) AND (("Glucocorticoids"[Mesh]) OR (“Cortico-steroids”[Mesh]))). Inclusion criteria encompassed reviews and case-control series; only articles written in English were included. Papers discussing different topics (bacterial encephalitis and autoimmune encephalitis) were excluded. Time or publication status restrictions were not applied. Additional studies were added based on a review of bibliographies of the identified papers.”

- ‘The main problem is that, at present, the paper does not provide helpful guidance to the clinician on the use of steroids in this context; in my opinion, it would be useful to better explore the role of steroids in the pathophysiology of these conditions and differentiating according to their nature (e.g. viral vs. parasitic, etc.); in particular, based on the mentioned trials, who are the patients who can benefit most from treatment? At what stage? For how long? What adverse effects have been observed? --> MODIFIED' Where have these suggested aspects been set out? --> ADDED: “Corticosteroids are strong anti-inflammatory agents able to reduce the secondary in-flammation-mediated damage in encephalitis of various nature, added to a specific an-tiviral treatment [31]. Beneficial effects of steroid therapy in viral encephalitis are an-ti-edematous action and possible prevention or therapy of secondary autoimmune phenomena [31]. Steroid use seems to be safe in most cases; no or only minor transient adverse effects are described [31].”

- ‘It would also be helpful to improve the discussion about the diagnosis to suggest at what stage it is useful/best to start steroid treatments;’ No response was provided to this comment; --> MODIFIED “Neuroimaging and lumbar puncture are essential initial diagnostic studies for evaluating patients with viral encephalitis. Lumbar puncture should be performed immediately in patients with suspected brain infection, and empirical treatment started immediately thereafter… Steroid therapy can be effective in the acute stage of viral encephalitis.”

- Similarly to this previous comment, ‘The Authors also correctly mention COVID-19-related forms of encephalitis. However, there are cases in the literature, especially in the context of predisposing factors such as thyroid disorders, in which cerebral venous thrombosis may also occur (see 10.3390/jpm13111557). These works should be cited in the discussion, and the clinical use of steroids in these contexts should be evaluated (haemorrhagic risk? Benefit on Sars-Cov2 infection and thyroid disorders? other relevant aspects?);' no answer was given, neither these aspects, important in my opinion in this context, have been integrated into the Discussion, whereas they could be very useful.  --> ADDED: “There are cases, especially in case of predisposing factors such as thyroid disorders, in which cerebral venous thrombosis may occur [32]. Corticosteroids have been associated with an increased risk of venous thromboembolism, but it is not clear whether the thrombotic risk is induced by the inflammation of the underlying inflammatory diseases or whether corticosteroids are also prothrombotic [33].”

I suggest the Authors complete their review by adding all these aspects and specifying where and how they made the recommended changes to the manuscript, and giving detailed explanations. 

Round 3

Reviewer 1 Report

Comments and Suggestions for Authors

This paper is a 2nd revised submission of a review of the use steroids in ‘non-bacterial and secondary encephalitis’. The authors have addressed some of the issues I had raised earlier, but a number still remain:

1.      Despite my raising this issue before, it is still uncertain when a new paragraph is being started – removing now all tabs has worsened the situation. The authors should have in fact have inserted tabs at the start of each new paragraph

2.      (major) Line 20 – the actual summary I had requested for before has still not been provided. Adding ‘bacterial and viral’ only worsens the situation as the paper is on non-bacterial?

3.      Line 51 – ‘in conjunction with immunoglobulin administration’ – this is untrue as steroids can be used in isolation without conjoint use with immunoglobulin

4.      Line 70 – with the now inclusion of only English publications, are there now fewer relevant papers?

Author Response

Dear reviewer,
here are the requested changes.

Sincerely,
the authors

  1. Despite my raising this issue before, it is still uncertain when a new paragraph is being started – removing now all tabs has worsened the situation. The authors should have in fact have inserted tabs at the start of each new paragraph --> ADDED.
  2. (major) Line 20 – the actual summary I had requested for before has still not been provided. Adding ‘bacterial and viral’ only worsens the situation as the paper is on non-bacterial? --> MODIFIED “Corticosteroid therapy represents a therapeutic option in the treatment of primary central nervous system diseases. Their use is also recommended in meningitis with autoimmune etiology”
  3. Line 51 – ‘in conjunction with immunoglobulin administration’ – this is untrue as steroids can be used in isolation without conjoint use with immunoglobulin --> MODIFIED: “Their use is also recommended in meningitis with autoimmune etiology [4], as first and second-line immunotherapy.
  4. Line 70 – with the now inclusion of only English publications, are there now fewer relevant papers? --> No, all articles were written in English.

Reviewer 4 Report

Comments and Suggestions for Authors

I thank the Authors for their work, which improved the quality of their paper. No further comments. 

Author Response

Dear reviewer,
thanks for the suggestions that allowed to improve the quality of the article.

Sincerely,
the authors